# Modulation of Vasomotor Function by Perivascular Adipose Tissue of Renal Artery Depends on Severity of Arterial Dysfunction to Nitric Oxide and Severity of Metabolic Parameters

**DOI:** 10.3390/biom12070870

**Published:** 2022-06-23

**Authors:** Satomi Kagota, Risa Futokoro, John J. McGuire, Kana Maruyama-Fumoto, Kazumasa Shinozuka

**Affiliations:** 1Department of Pharmacology, School of Pharmacy and Pharmaceutical Sciences, Mukogawa Women’s University, Nishinomiya 663 8179, Japan; futokoro@mukogawa-u.ac.jp (R.F.); k_maru@mukogawa-u.ac.jp (K.M.-F.); kazumasa@mukogawa-u.ac.jp (K.S.); 2Institute for Bioscience, Mukogawa Women’s University, Nishinomiya 663 8179, Japan; 3Departments of Medical Biophysics, Physiology & Pharmacology, Schulich School of Medicine & Dentistry, Western University, London, ON N6A 5C1, Canada; john.mcguire@schulich.uwo.ca

**Keywords:** adipose tissue, angiotensin II, metabolic syndrome, renal artery

## Abstract

Perivascular adipose tissue (PVAT) enhances vascular relaxation of mesenteric arteries in SHRSP.Z-*Lepr^fa^*/IzmDmcr rats (SPZF), a metabolic syndrome model. We investigated and compared the effects of PVAT on the renal artery in SPZF with those on SHR/NDmcr-cp rats (CP). Renal arteries with and without PVAT were isolated from 23-week-old SPZF and CP. The effects of PVAT on acetylcholine- and nitroprusside-induced relaxation were examined using bioassays with phenylephrine-contracted arterial rings. Acetylcholine-induced relaxations without PVAT in SPZF and CP were 0.7- and 0.5-times lower in females than in males, respectively. In the presence of PVAT, acetylcholine-induced relaxations increased 1.4- and 2-times in male and female CP, respectively, but did not differ in SPZF. Nitroprusside-induced relaxation with and without PVAT was 0.7-times lower in female than in male SPZF but did not differ in CP. Angiotensin-II type-1 receptor (AT1R)/AT1R-associated protein mRNA ratios were lower in CP than in the SPZF and negatively correlated with the difference in arterial relaxation with and without PVAT. The effects of renal artery PVAT differed between the SPZF and CP groups. Higher levels of enhanced AT1R activity in SPZF PVAT may drive these differences by impairing the vascular smooth muscle responses to nitric oxide.

## 1. Introduction

Perivascular adipose tissue (PVAT) regulates local blood vessel contractility by releasing vasoactive molecules. Adiponectin, hydrogen peroxide, hydrogen sulfide, and nitric oxide (NO) are PVAT-derived relaxing factors, while angiotensin II and reactive oxygen species are contracting factors [1,2,3,4]. The loss of the dilator effect in perivascular tissue is accompanied by an increase in the adipocyte area of obese patients [5]. The PVAT can play an important compensatory role when normal endothelial function is reduced. Dysfunction of PVAT’s effects revealed worsened underlying vascular dysfunction. Human clinical data provide evidence that PVAT surrounding the coronary arteries is associated with coronary inflammation and atherosclerosis and increases the risk of cardiovascular events [6]. Imaging perivascular fat using tomography angiography methodology, which can be used to monitor human coronary vessel inflammation, is proposed as a noninvasive method for detecting plaque instability, that is, before a serious event [7,8]. Thus, studying PVAT relaxation-enhancing mechanisms provides a better understanding of the significance of PVAT in the pathophysiology of vascular dysfunction in cardiovascular diseases [1,9,10].

SHRSP.Z-*Lepr^fa^*/IzmDmcr rats (SPZF), established by breeding stroke-prone spontaneously hypertensive rats with Zucker obese rats, are preclinical models of human hypertension with metabolic syndrome (MetS) [11,12]. Our previous work demonstrated that mesenteric arterial PVAT provides compensatory effects for impaired NO-dependent vasodilation in SPZF (17 and 20 weeks of age), but this compensatory system disappeared later in the time course of MetS (23 and 30 weeks of age) [13,14]. We have also recently found sex differences in the PVAT effects, in that the compensation for endothelium dysfunction extends to older age in females than in males in SPZF [15]. Furthermore, we showed that an increase in angiotensin II type a receptor (AT1R) or a decrease in AT1R-associated protein (ATRAP) expression in mesenteric arterial PVAT is associated with the deterioration of PVAT function in SPZF [14,15]. Overactivation of the systemic and local renin-angiotensin system (RAS) closely contributes to the onset of many diseases, including hypertension and cardiovascular disease. Inhibition of the RAS (e.g., with treatment by AT1R antagonists) is one of the most clinically useful and effective strategies to protect against the onset or aggravation of these diseases. The adipocyte RAS influences adipocyte growth and differentiation, regulates the production/release of adipokines, and promotes oxidative stress in adipocytes [4,16]. RAS overactivation in the PVAT contributes to vascular dysfunction in rats with heart failure [9]. ATRAP interacts with the carboxy-terminus of AT1R and inhibits AT1R signaling pathways that produce detrimental pathological effects via blood pressure regulation, cardiac and vascular hypertrophy, and adipose tissue metabolism [17,18,19]. Adipose tissue ATRAP plays an important role in preventing metabolic disorders by promoting adipogenesis and browning, protecting against insulin resistance, and improving adipose inflammation and function by suppressing the overactivation of adipose AT1R signaling [19,20,21]. AT1R/ATRAP ratios are increased in adipose tissue of spontaneously hypertensive rats (SHR) fed a high-fat diet [22]. Further, AT1R/ATRAP ratios were lowered by prehypertensive treatment with the AT1R blocker losartan, which also relieved the metabolic disorder in high-fat-diet-fed SHR [22]. Thus, adipose tissue RAS, that is, AT1R activity and its regulation by ATRAP, may serve as a molecular target for the treatment of metabolic disorders associated with visceral obesity.

A number of studies have described PVAT functions, but these descriptions have been heterogeneous. This may be explained, in part, by differences in study designs, such as species, age, sex, arterial sites, and metabolic status. For instance, renal perivascular adipose tissue, which is identified as a mix of brown and white adipocytes, differs from PVATs of the thoracic aorta and mesenteric artery in the anti-contractile response in male Sprague Dawley rats [23]. Obesity is a risk factor for the development of chronic kidney disease (CKD) [24]. Therefore, our current study using MetS models focuses on local PVAT modulation of renal arteries, which are large vessels that conduct blood directly from the abdominal aorta to the kidneys. We investigated the effects of changes in renal arterial PVAT on renal arterial tone and changes in mRNA expression of AT1R and ATRAP in the PVAT of SPZF. To test the translation of the findings between the models, we compared SPZF to age-matched SHR/NDmcr-cp rats (CP), another MetS model. CP is an inbred subline of SHR/N-corpulent rats that have a corpulent (cp) mutation of the leptin receptor (*Lepr*), an autosomal recessive trait that produces *Lepr* deficiency [25]. CP spontaneously develops hypertension with metabolic abnormalities, and an acetylcholine-induced NO response has been reported to be impaired in the thoracic aorta [26,27] and mesenteric arteries [28]. Furthermore, diabetic comorbidities, including cardiovascular and end-stage kidney disease, are more likely to occur in women than in men [29,30]. Similarly, CKD is more prevalent in women than in men, based on a meta-analysis using global population data [31]. We, therefore, investigated sex-related differences in renal PVAT effects, as has been reported for the mesenteric artery of SPZF. These sex-related differences in the PVAT may be clinically significant.

## 2. Materials and Methods

### 2.1. Experimental Animals

Male and female SPZF and CP, established by the Disease Model Cooperative Research Association (Kyoto, Japan), were purchased from Japan SLC, Inc. (Hamamatsu, Japan) and used at 23 weeks of age. The age range of animals is based on published aging time-course studies showing that the “breakdown” of the PVAT compensatory system was observed in the mesenteric arteries of male SPZF [13,14] but “preserved” in female SPZF [15]. All protocols involving the care and use of animals were approved by the Animal Ethics Committee and performed in accordance with the Guidelines for the Care and Use of Laboratory Animals at Mukogawa Women’s University (Protocol number: P-12-2020-01-A and P-12-2021-01-A).

### 2.2. Metabolic Parameters

One week before the date of the vascular function studies, systolic blood pressure (sBP) was assessed in each animal using the tail-cuff method (MK-2000; Muromachi, Tokyo, Japan), as described previously [32]. On the date of the study, body weight, waist circumference, and body length (nose to base of tail) were measured. The waist circumference-to-body length ratio of each rat was calculated as an index of abdominal obesity. At the same time of the day, non-fasting blood and urine samples were collected by needle puncture from the abdominal aorta and bladder of each rat under anesthesia with ketamine (90 mg/kg, i.p.) and xylazine (10 mg/kg, i.p.). The blood was centrifuged at 3000× *g* for 10 min at 4 °C (Allegra21R; Beckman Coulter Life Sciences, Tokyo, Japan), and serum levels of insulin and glucose were determined using commercial kits (catalog No. M1104 and 439-90901) from the Morinaga Biochemistry Lab. (Tokyo, Japan) and Wako Pure Chemical Industries. Ltd. (Osaka, Japan) and a Microplate Manager (model 680; Bio-Rad Laboratories, Inc., Tokyo, Japan). Protein levels were recorded using UROPAPER III ‘Eiken’ (E-UR80, Eiken Chemical Co., Ltd., Tochigi, Japan) on a 6-point scale (minus [none detected] = 0; 15 mg/100 mL = 1; 30 mg/100 mL = 2; 100 mg/100 mL = 3; 300 mg/100 mL = 4; 1000 mg/100 mL = 5), according to the manufacturer’s instructions. The glucose levels were also recorded using UROPAPER III ‘Eiken’ on a 5-point scale (minus (none detected) = 0; 50 mg/100 mL = 1; 100 mg/100 mL = 2; 250 mg/100 mL = 3; 500 mg/100 mL = 4). The hormonal differences between males and females were not considered in the present study.

### 2.3. Vascular Functions

Renal arteries were removed from the rats and immediately placed in Krebs–Henseleit (Krebs) buffer (pH 7.4; NaCl 118.4 mM, KCl 4.7 mM, MgSO_4_ 1.2 mM, CaCl_2_ 2.5 mM, NaHCO_3_ 25 mM, KH_2_PO_4_ 1.2 mM, and glucose 11.1 mM) saturated by bubbling with 95% O_2_ and 5% CO_2_ gas mixture (37 °C). Arteries with intact endothelium (with or without PVAT) were cut into 3 mm rings and mounted isometrically at an optimal resting tension (0.3 g) in 10 mL organ baths filled with Krebs solution. The vasomotor function of the arteries was determined using the organ bath method described previously [13]. Phenylephrine (0.1 nM–30 µM), acetylcholine (0.1 nM–1 µM), and sodium nitroprusside (0.1 nM–1 µM) were added cumulatively into the baths containing the arterial rings, and isometric tension changes were measured with a force-displacement transducer (Model t-7; NEC San-Ei, Tokyo, Japan) coupled to a dual-channel chart recorder (Model 8K21; NEC San-Ei). Arterial tings were submaximally contracted by phenylephrine to determine the acetylcholine and sodium nitroprusside dose–response curves. Individual concentration–response curves were analyzed using a nonlinear regression curve fitting of the relaxation–drug concentration relationships. Negative log EC_50_ and E_max_ were determined using GraphPad Prism 9 for macOS (ver. 9.3.1, San Diego, CA, USA).

### 2.4. Quantitative Real-Time PCR Assay

The mRNA transcript levels of AT1R and ATRAP in renal arterial PVAT were measured using quantitative real-time PCR assays, as previously described [33]. Total RNA was extracted from tissues and purified using the RNeasy fibrous tissue kit, according to the manufacturer’s instructions (Qiagen, Mississauga, Ontario, Canada), and real-time measurements of target gene expression were carried out using TaqMan RNA-to-CT 1-step kit and a LightCycler 1.5 (Roche Diagnostics Japan K.K., Tokyo, Japan). Commercially available gene-specific probes were used (Roche Applied Science, Universal ProbeLibrary product ID: AT1R, 04688503001; ATRAP, 04684982001; ribosomal protein 18S, 04688937001; β-glucuronidase, 04688015001; β-actin, 04686900001), and gene-specific primers designed by the Assay Design Center (Roche Applied Science) were purchased from Life Technologies Japan, Ltd. (Tokyo, Japan). A triad housekeeping gene expression approach (ribosomal protein 18S, β-glucuronidase, and β-actin) was used to normalize the sample material, and the efficiency of each primer set was included in all the calculations. The number of target genes was normalized to the reference genes to obtain the relative threshold cycle (ΔCT). They were then related to the CT levels of male rats to obtain the relative expression levels (2-ΔΔCT) of the target genes.

### 2.5. Data Analyses

Data are expressed as mean ± standard error of the mean (SEM). Statistical comparisons of means between groups were performed using a two-way analysis of variance (ANOVA). When ANOVA determined that the main effects (e.g., sex or PVAT state) were significant (*p* < 0.05), multiple comparisons were made using the Bonferroni post hoc test. Linear regression analysis was performed using GraphPad Prism. Statistical significance was set at *p* < 0.05.

### 2.6. Drugs

Additional reagents and chemicals were purchased from the following suppliers: L-phenylephrine hydrochloride (Sigma-Aldrich Co., LLC, St. Louis, MO, USA) and acetylcholine chloride (Daiichi Pharmaceutical Co., Ltd., Tokyo, Japan). Other chemicals of analytical reagent grade were purchased from Nacalai Tesque Inc. (Kyoto, Japan).

## 3. Results

### 3.1. Metabolic Parameters

As shown in Table 1, CP had heavier body weights than SPZF, and females of both strains had lower weights than males. Waist circumference–body length ratios and serum and urinary glucose levels did not differ among the four groups (Table 1). In both sexes, sBP was higher in the SPZF group than in the CP group (Table 1). In both strains, sBP levels were lower in females than in males. The serum insulin levels were higher in the SPZF group than in the CP group (Table 1). In the SPZF group, insulin levels were higher in females than in males. Urinary protein levels were altered by sex, and CP levels were lower in females than in males.

### 3.2. Changes in PVAT Response on Vasomotor Functions

Figure 1 and Table 2 show acetylcholine-induced endothelial NO-mediated relaxation (Figure 1a) and sodium nitroprusside-induced NO response in vascular smooth muscle cells (Figure 1b) in the renal arteries of SPZF. Acetylcholine- and nitroprusside-induced relaxation of arteries without PVAT were significantly less in females than in males and did not differ with the presence or absence of PVAT (Table 2).

In CP, the endothelial NO-mediated relaxation in females was smaller than those in males (shown in Figure 2a and Table 3), but there were no significant differences in the NO response in vascular smooth muscle cells between sexes (shown in Figure 2b and Table 3). Furthermore, acetylcholine-induced relaxation in the presence of PVAT significantly increased in both sexes (Figure 2a and Table 3).

Contractile responses to phenylephrine did not differ between the sexes or according to the presence of PVAT in either strain (Figure 3, Table 2 and Table 3).

### 3.3. mRNA Levels of AT1R and ATRAP in PVAT

The mRNA levels of AT1R and ATRAP in the renal arterial PVAT did not differ among the groups (Figure 4a,b). However, the AT1R/ATRAP ratio (Figure 4c), as an index of AT1R activity, was lower in CP than in SPZF and did not differ between sexes.

Furthermore, the AT1R/ATRAP ratio in PVAT was positively correlated with sBP (Figure 5a), while the ratio was negatively correlated with the differences in E_max_ values for acetylcholine in arteries with and without PVAT, as indices of the PVAT response to vascular relaxation (Figure 5b). No significant relationships were found between sBP or body weight and the PVAT response to vascular relaxation (*p* > 0.05). Serum insulin levels were also not significantly correlated with PVAT response to vascular relaxation (*p* > 0.05).

## 4. Discussion

The present study identified sex-related differences in acetylcholine- and nitroprusside-induced vasorelaxation of renal arteries in SPZF and CP rats. Furthermore, our study results indicate that PVAT function varies with the severity of the metabolic disorder, as shown by PVAT-enhancing relaxation in the renal arteries of CP, but not by age-matched SPZF. Interestingly, PVAT function did not differ between male and female CP. Lastly, a lower AT1R/ATRAP mRNA ratio in CP than in SPZF indicated lower renal adipose RAS activity and was correlated with the PVAT relaxation-enhancing effect in CP. Together, these findings suggest that PVAT plays a role in maintaining arterial function in the form of compensatory adjustments under conditions of impaired endothelial NO production. The results of this study highlight that AT1R activation is critical in the decline in PVAT function in vascular pathophysiology, suggesting that AT1R inhibition is beneficial for maintaining the compensatory PVAT effects.

Acetylcholine-induced relaxation is mediated by endothelial NO production in rat renal arteries from Sprague-Dawley rats [34], high-fat-fed rats [35], and Otsuka Long–Evans Tokushima Fatty rats, a model of diabetes mellitus [36]. In the current study, our findings indicate that acetylcholine-induced relaxation was lower in female CP and SPZF than in males. We suggest that these results are evidence of higher susceptibility to mechanisms that alter NO synthesis and signaling pathways in the renal arteries of females than in males. The coronary arteries of female ApoE KO mice fed a Western diet to induce atherosclerosis show a higher susceptibility to NO-dependent endothelial dysfunction [37]. In general, sex-related differences in cardiovascular diseases in human and animal models have been attributed to the protective mechanisms of estrogen. However, the impact of obesity on the development of cardiovascular disease appears to be greater in women than in men [38,39]. Women with ischemic heart disease have a higher mortality rate and more cardiovascular events than men [40]. Diabetes comorbidities, such as cardiovascular disease, heart failure, stroke, and kidney disease, are more likely to occur in women than in men [38]. Postmenopausal women have an increased risk of developing diabetic kidney disease [29]. The global prevalence of CKD is reported to be higher in women than in men [31]. Further studies of sex-related differences in renal arteries using preclinical models of MetS may provide insight into the mechanisms and approaches to address the higher risk of CKD in women.

This study showed that the effects of PVAT on the renal artery differed between the MetS models. Renal arterial PVAT enhanced relaxation, but this effect did not differ between sexes in CP at 23 weeks of age. In contrast, PVAT failed to enhance relaxation in age-matched SPZF. One explanation for these results may be the severity of the metabolic disorders in each model. Both SPZF and CP are leptin receptor-deficient rodent models [41], each of which carries a distinct genetic mutation; however, as shown in Table 1, their phenotypes differ with respect to physical characteristics and specific metabolic parameters. SPZF had a lower body weight but higher sBP and serum insulin levels compared to CP. However, these parameters were not directly correlated with the PVAT-enhancing effects on arterial tone. In contrast, the enhancing effect of PVAT on vasorelaxation was negatively related to the AT1R to ATRAP ratio in the renal arterial PVAT, which was positively related to sBP. These findings suggest that the difference in renal arterial PVAT response in the two MetS models may be due to AT1R activity, which may be explained by the increase in the AT1R to ATRAP ratio, in the PVAT. In other words, the renal–arterial PVAT modulation of renal arterial tone may be inhibited by the activation of angiotensin II signaling in PVAT, which is commonly elevated in hypertension. In recent studies on SPZF, we showed sex-related differences in the time-course of the decrease in mesenteric artery PVAT-induced relaxation that correlated with increased AT1R activity in mesenteric artery PVAT [14,15]. Inhibitory strategies targeting angiotensin II signaling in PVAT, such as treatment with AT1R blockers or ACE inhibitors, may be beneficial for the prevention and treatment of MetS-related organ injury. Prehypertensive losartan therapy relieved metabolic disorders by decreasing the AT1R to ATRAP ratio in adipose tissue in the later life of SHR fed a high-fat diet [22]. ATRAP is also expressed in renal tubules, where it regulates sodium handling by the kidneys [17]. Although urine protein was measured as an indicator of renal function in the present study, the determination of urinary sodium level may be valuable to the understanding of ATRAP function in renal PVAT. Downregulation of renal ATRAP is involved in the onset and progression of blood pressure elevation in a CKD model [42]. Regulation of AT1R activity via ATRAP in renal arterial PVAT, together with renal tubule ATRAP, may be important for maintaining normal kidney function. This concept may be supported by the results of the present study that under conditions of impaired endothelial NO-mediated vasorelaxation coincident with maintained PVAT-enhancing modulation, the urinary protein level of female CP was lower than in the others.

The present study has many limitations. Although the current study measured mRNA expression of AT1R and ATRAP in PVAT, it did not measure protein expression and its activity. We need to clarify whether enhanced AT1R activation occurs in renal arterial PVAT when the modulation effects are no longer present and whether treatment with AT1R blockers or ACE inhibitors can block the deterioration of the PVAT functions. In addition, other RAS components, including ACE2–Ang (1–7)–Mas receptor axis, play an important role in the vasodilatory pathway, and estrogen shifts the balance of the RAS towards the ACE2–Ang (1–7)–Mas receptor axis to protect against the development of cardiovascular diseases [43]. The relationship between changes in the expression and activity of the components in PVAT and the deterioration of PVAT modulation still needs to be clarified. In the current study, renal PVAT in both sexes failed to produce relaxation-enhancing effects, whereas mesenteric artery PVAT effects deteriorated in males [13,14] and remained in females [15] at 23 weeks of age. Taken together, these findings indicate site-dependent differences in the effects of PVAT on arterial tone. This hypothesis is supported by a previous report that the regulation of vascular tone by renal PVAT differs from thoracic aortas and mesenteric artery PVAT in male Sprague Dawley rats [23]. Arterial site-dependent differences in PVAT effects could imply different time courses for the deterioration, composition, or roles played by the function of the organ and tissues supplied by each artery. Further studies examining renal PVAT using younger SPZF may uncover sex-related differences, such as in mesenteric arteries. In addition, studies on renal PVAT using older CP will determine whether deterioration occurs at a later stage of MetS. Indeed, mesenteric arterial PVAT-enhancing effects also disappeared in female SPZF at 30 weeks of age [14,15]. The composition of PVAT could affect the factors produced at different arterial sites and the mechanisms affecting vascular function. In the current study, the factor(s) involved in PVAT-enhancing effects on renal arterial tone remained undetermined. Based on previous studies, we propose that apelin is a likely factor [14,15]. Apelin induces vasorelaxation via NO production from the vascular endothelium [44,45] and increases acetylcholine-induced relaxation of mesenteric arteries without PVAT in SPZF [14]. However, other factors, such as the gaseous transmitters CO and H_2_S, could be involved, given their interactions with the NO signaling pathways. Lastly, acetylcholine stimulates endothelium-derived hyperpolarizing factor(s) together with NO in the renal arteries of other rat strains [34,36]. Thus, PVAT effects and interactions with factors may be dependent on the composition of all endothelium-derived relaxing factors present within specific tissues.

## 5. Conclusions

The severity of metabolic disorders, high blood pressure, and vascular smooth muscle sensitivity to NO were factors affecting renal arterial PVAT vascular relaxation-enhancing effects of renal arterial PVAT in MetS models. PVAT activity did not differ between sexes; however, in the absence of PVAT, renal artery responses to endothelial agonists were lower in females than in males. This study adds to the accumulating evidence that negative regulation of AT1R signaling by ATRAP is a predictor of sustained favorable PVAT effects and highlights the importance of PVAT RAS overactivation in vascular dysfunction and MetS.

## Figures and Tables

**Figure 1 biomolecules-12-00870-f001:**
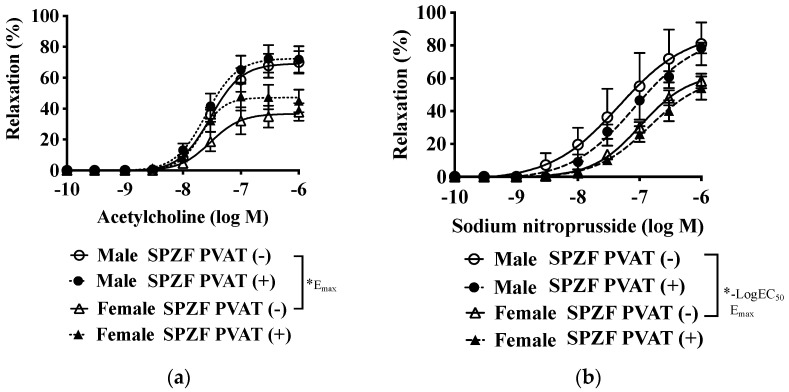
Vascular relaxation in response to acetylcholine (**a**) and sodium nitroprusside (**b**) in renal arteries with or without perivascular adipose tissues [PVAT (+/−)] of male and female SHRSP.Z-*Lepr^fa^*/IzmDmcr rats (SPZF) at 23 weeks of age. Arterial rings were submaximally contracted by phenylephrine to determine the acetylcholine and sodium nitroprusside dose–response curves. Data are expressed as means ± SEM. *n* = 9 rats/group. * *p* < 0.05. Statistical comparisons of means between groups were performed using a two-way ANOVA followed by Bonferroni post-hoc test.

**Figure 2 biomolecules-12-00870-f002:**
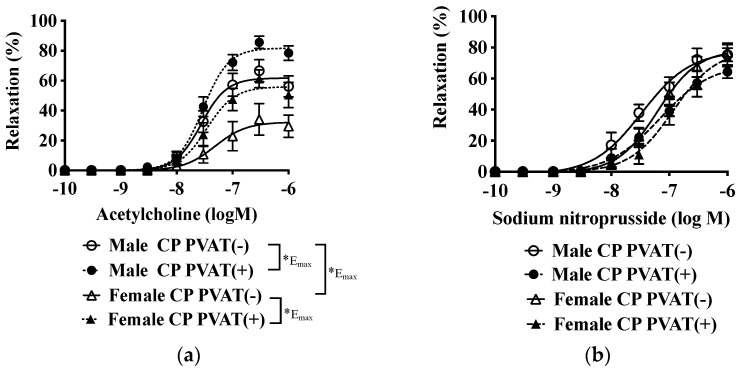
Vascular relaxation in response to acetylcholine (**a**) and sodium nitroprusside (**b**) in renal arteries with or without perivascular adipose tissues [PVAT (+/−)] of male and female SHR/NDmcr-cp rats (CP) at 23 weeks of age. Arterial rings were submaximally contracted by phenylephrine to determine the acetylcholine and sodium nitroprusside dose–response curves. Data are expressed as means ± SEM. *n* = 9 rats/group. * *p* < 0.05. Statistical comparisons of means between groups were performed using a two-way ANOVA followed by Bonferroni post hoc test.

**Figure 3 biomolecules-12-00870-f003:**
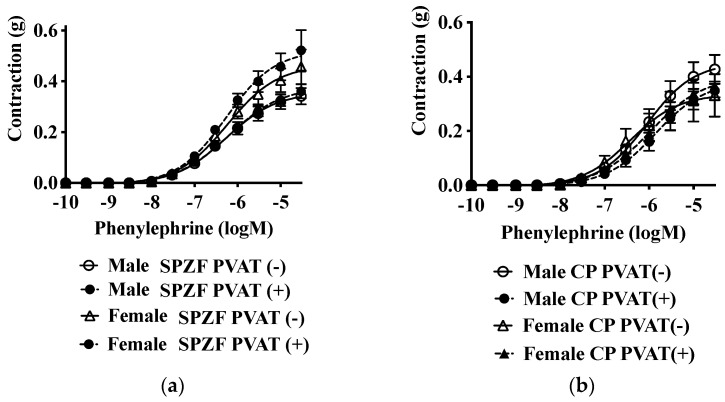
Vascular contractile response to phenylephrine in renal arteries with or without perivascular adipose tissues [PVAT (+/−)] of male and female SHRSP.Z-*Lepr^fa^*/IzmDmcr rats (SPZF, **a**) and SHR/NDmcr-cp rats (CP, **b**) at 23 weeks of age. Data are expressed as means ±SEM. *n* = 9 rats/group.

**Figure 4 biomolecules-12-00870-f004:**
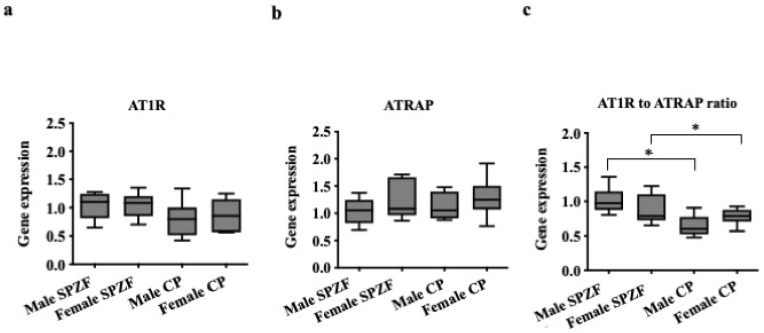
mRNA expression of angiotensin II type 1a receptor (AT1R, **a**), AT1R-associated protein (ATRAP, **b**), and AT1R to ATRAP ratio (**c**) in renal arterial perivascular adipose tissues (PVAT) of male and female SHRSP.Z-*Lepr^fa^*/IzmDmcr rats (SPZF) and SHR/NDmcr-cp rats (CP) at 23 weeks of age. * *p* < 0.05. Data are expressed as means ± SEM. *n* = 9 rats/group. * *p* < 0.05. Statistical comparisons of means between groups were performed using a two-way ANOVA followed by Bonferroni post hoc test.

**Figure 5 biomolecules-12-00870-f005:**
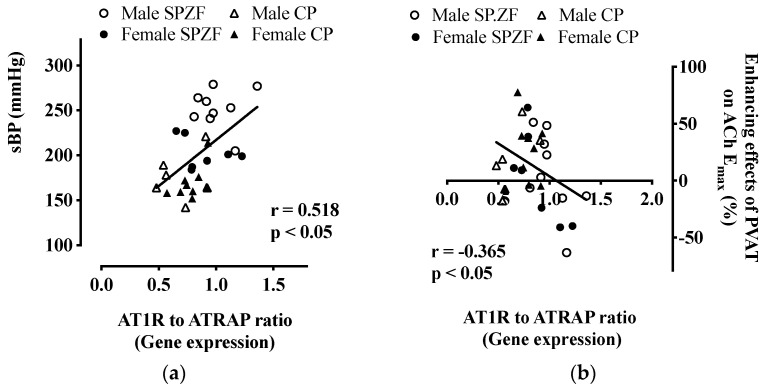
Correlations between systolic blood pressure and mRNA expression of angiotensin II type 1 receptor (AT1R) to AT1R-associated protein (ATRAP) ratio (**a**), and the AT1R to ATRAP ratio and enhancement of acetylcholine-induced relaxation by perivascular adipose tissues (PVAT) (**b**) in male and female SHRSP.Z-*Lepr^fa^*/IzmDmcr rats (SHRSP.ZF) and SHR/NDmcr-cp rats (CP) at 23 weeks of age. Relaxation enhancement was assessed by differences in maximum response to acetylcholine with or without PVAT. Data are expressed as means ± SEM. *n* = 9 rats/group.

**Table 1 biomolecules-12-00870-t001:** Characteristics of SHRSP.Z-*Lepr^fa^*/IzmDmcr rats (SPZF) and SHR/NDmcr-cp rats (CP).

Group (*n*)	SPZF		CP	
	Male	Female	Male	Female
Body weight (g)	493 ± 7	447 ± 5 ^b,^*	582 ± 13 ^a^	469 ± 12 ^a,b,^*
Waist circumference–body length ratio (cm/cm)	1.04 ± 0.01	1.09 ± 0.01	1.03 ± 0.01	1.05 ± 0.02
sBP (mmHg)	246 ± 7	200 ± 5 ^b,^*	176 ± 11 ^a^	169 ± 6 ^a,b,^*
Serum levels				
Glucose (mg/100 mL)	356 ± 25	261 ± 28	321 ± 38	357 ± 35
Insulin (ng/mL)	37.1 ± 4.9	54.8 ± 7.0 *	30.4 ± 6.3 ^a^	20.1 ± 4.3 ^a^
Urine levels				
Glucose (score)	0.18 ± 0.18	0.63 ± 0.24	0.08 ± 0.07	0.14 ± 0.13
Protein (score)	3.27 ± 0.14	3.78 ± 0.15 ^b^	3.30 ± 0.21	2.23 ± 0.20 ^b,^*

Results are expressed as the mean ± SEM of 9 rats/group. Statistical comparisons of parameters were performed using a two-way (sex × animal model) analysis of variance (ANOVA) followed by Bonferroni post hoc test. ^a^
*p* < 0.05, main effect of the animal model; ^b^
*p* < 0.05, main effect of sex; * *p* < 0.05, male vs. female (Bonferroni).

**Table 2 biomolecules-12-00870-t002:** Relaxation and contractile response in renal arteries with (+) or without (−) perivascular adipose tissue (PVAT) from SHRSP.Z-*Lepr^fa^*/IzmDmcr rats (SPZF).

Agonist and Activity	CRC Parameter	PVAT State	Sex	
			Male	Female
Acetylcholine-induced relaxation	−Log EC_50_	PVAT (−)	7.53 ± 0.11	7.29 ± 0.15
		PVAT (+)	7.70 ± 0.17	7.61 ± 0.09
	E_max_ (%)	PVAT (−)	65.8 ± 7.2 ^b^	39.4 ± 6.9 ^b^
		PVAT (+)	75.1 ± 7.4	50.3 ± 8.7
Sodium nitroprusside-induced relaxation	−Log EC_50_	PVAT (−)	7.16 ± 0.17 ^b^	6.92 ± 0.08 ^b^
		PVAT (+)	6.98 ± 0.14	6.86 ± 0.06
	E_max_ (%)	PVAT (−)	93.2 ± 3.1 ^b^	66.1 ± 5.4 ^b^
		PVAT (+)	99.0 ± 6.5	66.9 ± 7.2
Phenylephrine-induced contraction	−Log EC_50_	PVAT (−)	6.33 ± 0.09	6.44 ± 0.09
		PVAT (+)	6.27 ± 0.12	6.50 ± 0.08
	E_max_ (g)	PVAT (−)	0.487 ± 0.047	0.497 ± 0.078
		PVAT (+)	0.485 ± 0.020	0.425 ± 0.078

Results are expressed as the mean ± SEM of 9 rats/group. Statistical comparisons of each concentration–response curve (CRC) parameters obtained in renal arteries from males and females of SPZF were performed using a two-way (sex × PVAT state) analysis of variance (ANOVA) followed by Bonferroni post hoc test. ^b^
*p* < 0.05, main effect of sex (Bonferroni).

**Table 3 biomolecules-12-00870-t003:** Relaxation and contractile response in renal arteries with (+) or without (−) perivascular adipose tissue (PVAT) from SHR/NDmcr-cp rats (CP).

Agonist and Activity	CRC Parameter	PVAT State	Sex	
			Male	Female
Acetylcholine-induced relaxation	−Log EC_50_	PVAT (−)	7.38 ± 0.11	7.27 ± 0.20
		PVAT (+)	7.48 ± 0.09	7.41 ± 0.12
	E_max_ (%)	PVAT (−)	58.7 ± 5.9 ^a,b^	29.2 ± 8.6 ^a,b^
		PVAT (+)	81.4 ± 5.4 *	57.4 ± 6.3 *
Sodium nitroprusside-induced relaxation	−Log EC_50_	PVAT (−)	7.43 ± 0.13	7.18 ± 0.04
		PVAT (+)	7.15 ± 0.09	6.88 ± 0.16
	E_max_ (%)	PVAT (−)	80.9 ± 5.4	79.7 ± 4.7
		PVAT (+)	72.6 ± 3.8	83.4 ± 11.1
Phenylephrine-induced contraction	−Log EC_50_	PVAT (−)	6.11 ± 0.10	5.95 ± 0.32
		PVAT (+)	5.38 ± 0.55	5.93 ± 0.07
	E_max_ (g)	PVAT (−)	0.508 ± 0.062	0.363 ± 0.070
		PVAT (+)	0.506 ± 0.118	0.404 ± 0.041

Results are expressed as the mean ± SEM of 9 rats/group. Statistical comparisons of each concentration–response curve (CRC) parameters obtained in renal arteries from males and females of CP were performed using a two-way (sex × PVAT state) analysis of variance (ANOVA) followed by Bonferroni post hoc test. ^a^
*p* < 0.05, main effect of the presence of PVAT; ^b^
*p* < 0.05, main effect of sex; ^*^
*p* < 0.05, PVAT (+) vs. PVAT (−) (Bonferroni).

## Data Availability

The data presented in this study are available upon request from the corresponding author.

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
