# Peer review of "Modulation of Vasomotor Function by Perivascular Adipose Tissue of Renal Artery Depends on Severity of Arterial Dysfunction to Nitric Oxide and Severity of Metabolic Parameters"

_biomolecules, 2022, doi:10.3390/biom12070870_

Round 1

Reviewer 1 Report

After evaluating the manuscript titled "Modulation of vasomotor function by perivascular adipose tissue of renal artery depends on severity of arterial dysfunction to nitric oxide and severity of metabolic parameters," I recommend the authors address the following:

Introduction

1)    Do the authors mean that the authors cite in these references 1-4 (lines 30-32) identified Ang-II as a PVAT-derived contracting factor?

2)    The last paragraph of the introduction needs some work. For example, perhaps, the passage in lines 89-92 (“Diabetic comorbidities, including cardiovascular disease and end-stage kidney disease, are more likely to occur in women than in men [28,29]. Similarly, CKD is more prevalent in women than in men, based on meta-analyses using global population data”), could come before the sentences in lines 87-89.

3)    The sentence in line 87 needs attention regarding the English style.

4)    The whole text will be enhanced after a careful edition of the English style and typos.

Methods

11)    How did the authors determine the differential (+PVAT and – PVAT) optimal resting tension (0.3 g) in a renal artery?

22)    Did the authors consider the hormonal differences between males and females in the protocols and data analysis? Clarify it, please.

33)    Which approach did the authors use to precontract the arteries to execute the concentration-response curves for ACh and NO? Please describe it in the methods section and the respective legends.

44)    When proceeding with the functional studies, did the authors work with renal arteries containing intact endothelium in all groups (ACh, SNP, and phenylephrine)? If not, please explain why you did not remove the endothelium and describe how you tested and assured the presence of endothelium. If you removed the endothelial layer, please explain why you did it and how you removed it. Please describe how you confirmed the PVAT was not damaged during the endothelium removal if that is the case.)    Did the authors measure the sodium levels or sodium excretion? Wouldn’t it be relevant to reinforce the discussion found in the lines 292-293: “ATRAP is also expressed in renal tubules, where it regulates sodium handling by the kidneys

Discussion

1) Since the authors discuss “endothelium-derived relaxing factors” (e.g., in lines 319-321), please consider discussing the influence of endothelial cells mentioning if/how/why you keep (or not) the endothelial cells (consideration about endothelial removal/maintained must have been described in the methods – as recommended above).

2) Authors should clarify that studies using ATR1 antagonists were not done, which may limit their conclusions. Why did the authors not draw considerations about the ACE inhibition as well as they did for RAT1 inhibition?

3) A limitation section is missing.

4) Which factors, besides estrogen, would contribute to the CP and SPZF females presenting with a higher susceptibility to mechanisms that alter NO synthesis and signaling pathways in the renal arteries than in males (to provide considerations on this questions, it is important to describe to the reader if/how/why (or why not) the authors measured estrogen or made other approaches to collect the baseline hormones levels among the females to have standardized data in this cohort).

5) The authors mentioned (lines 282-283) that “These findings suggest that the difference in renal arterial PVAT response in the two MetS models may be due to AT1R activity in the PVAT”. Did the authors measure the AT1R activity in other studies with the exact same protocol? Clarify it.

6) To reinforce the statement in lines 302-302 (“Taken together, these findings indicate site-dependent differences in the effects of PVAT on arterial tone), I suggest considering the following manuscript: https://pubmed.ncbi.nlm.nih.gov/29454047/

This manuscript may also provide support for statements in the introduction, e.g., in lines 74-76 (“Across studies, PVAT functions have been heterogeneously described, which may be explained in part by differences in study designs, such as species, age, sex, arterial sites, and metabolic status.)

Author Response

We thank you and the reviewer for the thoughtful suggestions that have increased the merits of this manuscript and study. Our responses to the comments are provided below.

Introduction

1) Yes. References 2 and 4 say Ang II is a PVAT-derived contractile factor. 

2) According to your suggestion, the sentences in lines 89-92 moved before the sentences in lines 87-89.
Page 2, Lines 99-104. “Furthermore, diabetic comorbidities, including cardiovascular and end-stage kidney disease, are more likely to occur in women than in men [29,30]. Similarly, CKD is more prevalent in women than in men, based on a meta-analysis using global population data [31]. We, therefore, investigated sex-related differences in renal PVAT effects, as has been reported for the mesenteric artery of SPZF. These sex-related differences in the PVAT may be clinically significant.”

3) We use a professional English editing service, “Editage,” to improve our English and correct typographical errors. We requested that they carefully edit the manuscript again.

4) We reported your suggestion to our editing service to address these errors.

Methods

11) According to a report from Watts et al. (Vascular Pharmacology, 2018, 106, 37), doing the step-wise, cumulative length-tension response curves in the renal artery with/without PVAT is the best way to determine the differential optimal resting tension in a renal artery. Regardless of the presence or absence of PVAT, we have used 0.3 g as the resting tension for the renal artery to determine vascular function in accordance with our previous studies. We have stated this in the Results.
Lines 161-163: “Arteries with intact endothelium (with or without PVAT) were cut into 3-mm rings and mounted isometrically at an optimal resting tension (0.3 g) in 10-mL organ baths filled with Krebs solution.”

22) Thank you for your question. We did not consider the hormonal differences in this study.  We have added a sentence to the Methods section where we state this. Furthermore, according to another reviewer's suggestion, we added the information on the role of the ACE2/Ang (1-7)/Mas receptor system on vasomotor function and the relation between the system and estrogen in the limitations section of the Discussion.
Lines 155-156: “The hormonal differences between males and females were not considered in the present study.”
Lines 388-391: “In addition, other RAS components, including ACE2/Ang (1-7)/Mas receptor axis, play an important role in the vasodilatory pathway, and estrogen shifts the balance of the RAS towards the ACE2/Ang (1-7)/Mas receptor axis to protect against the development of cardiovascular diseases [43].”

33) Thank you for drawing our attention to this omission. We explained how to measure the concentration-response curves for acetylcholine and sodium nitroprusside in the Methods and the respective legends.
Lines 169-170; 236-237; 255-256: “Arterial rings were submaximally contracted by phenylephrine to determine the acetylcholine and sodium nitroprusside dose response curves.”

44) We used only endothelium intact renal arterial ring preparations. We stated that renal arteries with intact endothelium were used in the Methods. We also stated we did not measure the sodium levels in urine in the present study, and the determination may be valuable for understanding ATRAP function in PVAT together with kidneys in the Discussion.
Lines 161-163: “Arteries with intact endothelium (with or without PVAT) were cut into 3-mm rings and mounted isometrically at an optimal resting tension (0.3 g) in 10-mL organ baths filled with Krebs solution.”
Lines:351-354: “Although urine protein was measured as an indicator of renal function in the present study, the determination of urinary sodium level may be valuable to the understanding of ATRAP function in renal PVAT.”

Discussion

1) As we mentioned above, we stated that we used the renal arteries with intact endothelium in the Methods.

2) According to your and another reviewer’s suggestions, we stated some limitations of the present study, including the effects of treatment with AT1R antagonists or ACE inhibitors on PVAT modifications on vasomotor functions, in the last section of the Discussion.
Lines 383-393: “The present study has many limitations. Although the current study measured mRNA expression of AT1R and ATRAP in PVAT, it did not measure protein expressions and its activity. We need to clarify whether enhanced AT1R activation occurs in renal arterial PVAT when the modulation effects are no longer present, and whether treatment with AT1R blockers or ACE inhibitors can block the deterioration of the PVAT functions. In addition, other RAS components, including ACE2/Ang (1-7)/Mas receptor axis, play an important role in the vasodilatory pathway, and estrogen shifts the balance of the RAS towards the ACE2/Ang (1-7)/Mas receptor axis to protect against the development of cardiovascular diseases [43]. The relationship between changes in the expression and activity of the components in PVAT and the deterioration of PVAT modulation still needs to be clarified.”

3) We have added a “limitation” section to the Discussion to state the limitations of the present study (Lines 383-416).

4) We added the information on hormones including estrogen as the limitations of the present study in the Discussion.
Lines 388-391: “In addition, other RAS components, including ACE2/Ang (1-7)/Mas receptor axis, play an important role in the vasodilatory pathway, and estrogen shifts the balance of the RAS towards the ACE2/Ang (1-7)/Mas receptor axis to protect against the development of cardiovascular diseases [43].”

5) We stated that the current study did not measure AT1R activity directly. We noticed it and explained why we speculated an increase AT1R activity in the Discussion. 
Lines 339-341: “These findings suggest that the difference in renal arterial PVAT response in the two MetS models may be due to AT1R activity, which may be explained by the increase in the AT1R to ATRAP ratio, in the PVAT.”

Lines 383-387: " Although the current study measured mRNA expression of AT1R and ATRAP in PVAT, it did not measure protein expressions and its activity. We need to clarify whether enhanced AT1R activation occurs in renal arterial PVAT when the modulation effects are no longer present, and whether treatment with AT1R blockers or ACE inhibitors can block the deterioration of the PVAT functions. “

6) Thank you for the suggestion. We have added the manuscript to our References (Ref. 23) and includes some sentences into the Introduction and Discussion referring to the manuscript.
Lines 85-88: “For instance, renal perivascular adipose tissue, which is identified as a mix of brown and white adipocytes, differs from PVATs of the thoracic aorta and mesenteric artery in the anti-contractile response in male Sprague Dawley rats [23].”
Lines 396-398: “This hypothesis is supported by a previous report that the regulation of vascular tone by renal PVAT differs from thoracic aorta and mesenteric artery PVAT in male Sprague Dawley rats [23].”

Reviewer 2 Report

Dear Authors,

The article is well written. The experiment is well designed, the discussion is sufficient and the conclusions are well constructed. The only thing which can be improved in my opinion is to provide "limitations" section. It is not obligatory in Biomolecules journal but in my opinion, it would increase the value of the article.

Thank you

Author Response

We thank you and the reviewer for the thoughtful suggestions that have increased the merits of this manuscript and study. Our responses to the comments are provided below.

We have added a “limitation” section to the Discussion to state the limitations of the present study (Lines 383-416).

Reviewer 3 Report

This is an interesting research article investigating the impact of adiposity on vascular tone and the associated dysfunction and functional alteration in the context of metabolism and renin angiotensin system (RAS). RAS per se has a broader impact in the vascular as well as metabolic function, which authors have touched up on in a subtle fashion which could be fixed, it undercuts the storyline; otherwise this work brings newer information to the field. Here are few comments - 

1. Sex is always a bigger variation in the field of biology, especially in the context of RAS signals executing metabolic and vascular function because ACE2, one of the RAS components is expressed in the X chromosome. Because authors have explored about the gene expression of AT1R, the story goes incomplete without ACE and ACE2. 

2. Authors are required to do a gene expression assessment of other RAS components such as ACE and ACE2 and extrapolate the story with vascular tone in addition to the existing one. Authors could brief about ACE/ACE2, its genetic location and role in vascular function in the introduction too. Alternatively, they could measure the ACE/ACE2 levels in the plasma too. 

Author Response

We thank you and the reviewer for the thoughtful suggestions that have increased the merits of this manuscript and study. Our responses (bold font) to the comments are provided below.

  1. As you suggested, ACE2/Ang (1-7)/Mas receptor axis is an important system that regulates vascular tone, and it has been reported that estrogen shifts the balance of the RAS towards ACE2/Ang (1-7)/Mas receptor axis to protect against the development of cardiovascular diseases. In referring to a recent review report (Ref. 43), we stated the importance of investigating the role of the ACE2/Ang (1-7)/Mas receptor system and its relation to estrogen on vasomotor regulation by PVAT in the limitation section of the Discussions.
    Lines 388-393: “In addition, other RAS components, including ACE2/Ang (1-7)/Mas receptor axis, play an important role in the vasodilatory pathway, and estrogen shifts the balance of the RAS towards the ACE2/Ang (1-7)/Mas receptor axis to protect against the development of cardiovascular diseases [43]. The relationship between changes in the expression and activity of the components in PVAT and the deterioration of PVAT modulation still needs to be clarified.”
  2. Thank you for the suggestion. We will do the experiments to find the role of the ACE2/Ang (1-7)/Mas receptor system, together with ACE/Ang II/AT1 receptor system, on the regulation by PVAT on vascular tone in future studies. 
